# A Study of Wave Confinement and Optical Force in Polydimethlysiloxane–Arylazopyrazole Composite for Photonic Applications

**DOI:** 10.3390/polym14050896

**Published:** 2022-02-24

**Authors:** Ikemefuna Uba, Demetris Geddis, Kesete Ghebreyessus, Uwe Hömmerich, Jerald Dumas

**Affiliations:** 1Department of Electrical and Computer Engineering, Hampton University, Hampton, VA 23668, USA; ikemefuna.uba@hamptonu.edu; 2Department of Chemistry and Biochemistry, Hampton University, Hampton, VA 23668, USA; kesete.ghebreyessus@hamptonu.edu; 3Department of Physics, Hampton University, Hampton, VA 23668, USA; uwe.hommerich@hamptonu.edu; 4Department of Chemical Engineering, Hampton University, Hampton, VA 23668, USA; jerald.dumas@hamptonu.edu

**Keywords:** arylazopyrazole, polydimethylsiloxane, waveguide, optical force

## Abstract

A refractive index of dielectrics was modified by several methods and was known to have direct influence on optical forces in nanophotonic structures. The present contribution shows that isomerization of photoswitching molecules can be used to regulate refractive index of dielectrics in-situ. In particular, spectroscopic study of a polydimethylsiloxane–arylazopyrazole (PDMS–AAP) composite revealed that refractive index of the composite shifts from 2.0 to 1.65 in trans and cis states, respectively, of the embedded AAP. Based on this, a proposition is made for a waveguide structure, in which external UV/Vis source reversibly regulates the conformation of the PDMS–AAP core. Computational study is performed using Maxwell’s equations on buried waveguide structure. The simulation, implemented in PYTHON, sequentially utilizes empirical refractive indices of the composite in the isomeric states in lieu of regulation by a source. The simulation revealed highly confined wave propagations for injected signals of 340 and 450 nm wavelengths. It is observed that the cis state suppresses higher order mode when propagating UV wavelength but allows it for visible light. This modal tuning demonstrated that single mode can be selectively excited with appropriate waveguide dimensions. Further impact of the tuning is seen in the optical force between waveguide pair where the forces shift between attractive and repulsive in relation to the isomeric state of the PDMS–AAP core. These effects which stem from the adjustment of refractive index by photoisomerization suggests that in-situ regulation of index is achievable by successful integration of photoswitching molecules in host materials, and the current PDMS–AAP composites investigated in this study can potentially enhance nanophotonic and opto-mechanical platforms.

## 1. Introduction

The progress of modern technology rests exclusively on the properties of artificial materials. The interaction of light with electronic and mechanical characteristics of such materials pushes efforts in guided optics, optoelectronics, and optofluidic platforms. Light-related phenomena in materials such as bandgap, photoconductivity, wave confinement, etc., very well depend on the refractive index, such that control of the latter is one means to modify phenomena of interest. Moreover, correlation of refractive index and frequencies has a general interpretation in the Kramer–Kronig (KK) relation [1]. In a recent study of hemoglobin solution where the refractive index depended on hemoglobin concentration, it was demonstrated that KK relation can be transformed in terms of wavelength [2], while Kim et.al showed that the relation is also differentiable, which is suitable for practical measurement without the knowledge of the imaginary part of the index [3]. These reports imply that refractive index of a material for a set range of wavelengths can be interpreted by a power series, whose coefficients consist of derivatives of the KK relations to accommodate nonmonotonic variations that may exist. Enhancement of index is important in nanophotonics, where wave confinement is central and revived interest in optical forces between photonic channels [4,5]. Due to their pivotal role in photonic devices, polymeric materials, besides other dielectrics, draw efforts in the manipulation of refractive indices. Adjustments were achieved via introduction of co-polymers, organic rings, and metal and semiconducting nanoparticles [6,7,8,9]. Also, at synthesis level, index modification was attained by dopant-controlled polymerization [10,11]. Consequently, there is simultaneous investigation of waveguide structures of these materials [12,13] and waveguide–waveguide or waveguide–substrate interaction forces [14,15]. As demonstrated by Povinelli [16], the force can be positive (repulsive) or negative (attractive), and this is due to evanescent coupling, i.e., by the attenuated wave in the clad. Thus, phenomenological description of the force can be in the form, $F\left(x\right)=\pm F\left(0\right)e^{-\sigma x},$ corresponding to zero evanescent coupling wave when the separation x → ∞. F(0) is the value at zero separation of the structures, and σ is a constant. Rigorous considerations showed this force to depend on refractive index, while the attribute and strength can be tuned respectively by phase difference and mode of waves propagating in the channels [4,17]. Since refractive index is crucial to the interaction force, it makes sense to consider that if index can be modified in-situ, it potentially can provide another means to control the force and different kinds of applications. One way to explore this is by introducing photoswitching molecules into the primary dielectric host that constitute, for example, the waveguide core.

Photoswitchable molecules are organic chains that reversibly change chemical structure from trans-to-cis on exposure to lights of certain wavelengths. They were used to achieve mechanical actuation and energy storage [18,19,20]. If successfully integrated into a host that promises stoichiometric freedom, not only can a photoswitch induce a new set of indices in the material, but it can also reversibly switch the value through photoisomerization. Among several photoswitches, aryazopyrazole (AAP) was successfully embedded into polydimethylsiloxane (PDMS) [21,22]. The PDMS–AAP composite exhibited actuation, high index (≈2.0), and regulation of same via isomerization. We note that AAPs have thermal stability and half-life of 1000 days, unlike ubiquitous azobenzene, and they attain high switching efficiency [23]. In particular, the AAP derivative reported in [21,22] achieved ≈ 100% efficiency in solid matrix over repeated switching. It is then logical to examine suitability of this composite for nanophotonic circuits.

In this paper, we propose a waveguide consisting of PDMS–AAP core where the index may be regulated in-situ by alternative exposure to UV and visible light. We utilized buried geometry in which the core is surrounded by PDMS clad known to be transparent to UV and visible lights. A semiempirical simulation of electric field, in transverse electric field (TE) mode, of a signal injected into the core is carried out via a program to examine wave confinement and the effect of changing the index in-place. The practical alternating exposure of the core to UV/Visible source is ensured by sequential implementation of the empirical refractive indices of the core materials in the solutions to Maxwell’s equations for invariant design parameters. In addition, optical force between a pair of such guides is investigated in relation to the isomeric states of the embedded AAP. In what follows, we describe the simulations that are based on existing literature and analyze the results for practical feasibility.

## 2. Materials and Methods

### 2.1. Materials 

Commercially available reagents were utilized for the synthesis of the AAP photomolecule and PDMS–AAP composite films. N-methylhydrazine, 1-bromo-6-hexanol, ethyl acetate, Na_2_SO_4_, K_2_CO_3_, acetone, potassium iodide, 4-aminophenol, 2,4-pentanedione, dichloromethane, and polydimethylsiloxane (elastomer and curer) were procured from Sigma–Aldrich, St. Louis, MO, USA and used as received.

### 2.2. Characterization 

The optical characterizations were carried out using a Shimazu 3600 spectrophotometer (Tokyo, Japan) in 300–800 nm. Light-induced reversible switching was performed by irradiating with 365 nm UV light for *trans*-to-*cis* isomerization (trans state), and green light 525 nm *cis*-to-*trans* isomerization (cis state). 

### 2.3. Synthesis of Molecular Switch 

The arylazopyrazole (AAP) molecular switch was synthesized following procedures developed in our previous studies using commercially available aniline [21,22]. 

### 2.4. Synthesis of PDMS–AAP Composite

Polydimethylsiloxane–arylazopyrazole (PDMS–AAP) composite was fabricated by spin-coating PDMS elastomer-curer gel doped with 0.02M of AAP on glass substrate and curing at 150 °C for 22 min as described in our previous work [21]. 

## 3. Results and Discussions

### 3.1. Photoisomerization Properties 

The photoisomerization behavior of the pure AAP and PDMS–AAP composite was investigated by UV/Vis spectroscopy with regard to switching behavior and absorbance maxima of the *trans* and *cis*-isomers. Both the pure AAP molecular switch and the PDMS–AAP composite showed distinct behavior upon irradiation with UV light (λ = 365 nm) for the *trans*-to-*cis* isomerization (trans state) and green light (λ = 525 nm) for the *cis*-to-*trans* isomerization (cis state). An irradiation with green light of the PDMS–AAP composite resulted in the original absorbance spectrum of the trans state being retrieved with almost 100% of the original intensity. This distinct behavior is due to the efficient reversible isomerization the AAP unit embedded in the PDMS matrix [21,22]. Comparison of the index of refraction in the trans and cis states as well as that of pure PDMS of the same thickness were described in our recently reported studies [21]. 

### 3.2. Refractive Index of PDMS–AAP

Refractive indices of the PDMS–AAP composite in comparison with pure PDMS is described in Figure 1 with a significant response to the structural state of embedded AAP. The indices were determined from the well-known relation $R={\left(n-1\right)}^{2}+\kappa^{2}/{\left(n+1\right)}^{2}+\kappa^{2}$. Where R is reflectance and *κ* is extinction coefficient. The *trans* state (Figure 1a) caused index maxima of 2.00 at 340 nm while the *cis* state (Figure 1b) caused a drop to 1.65 at this wavelength and a maximum at 450 nm with index of 1.67.

In both cases, the presence of localized maxima puts the index–wavelength response outside the domain of common index models (Sellmeier, Cauchy, etc.) that rather describe monotonic decreasing response. Therefore, we interpret the index–wavelength response with a Taylor series center on lowest wavelength, λ_1_:(1)$$n\left(\lambda_{1}\right)+\overset{N}{\underset{s=1}{\sum }}\frac{f^{\left(s\right)}\left(\lambda_{1}\right)}{s!}{\left(\lambda -\lambda_{1}\right)}^{s}$$
where $f^{\left(s\right)}\left(\lambda_{1}\right)$ is s-order derivative of KK relation in wavelength range outside of the singularity point. We found a six-term series to be a good fit with the composite index shown in Figure 1. Scaling factors can be introduced to the coefficients of Equation (1) to reduce the number of terms needed for convergence without violating convergence conditions. Figure 1 shows that the indices of PDMS–AAP composite are well above that of pure PDMS at both isomeric states. One of the implications is that a waveguide structure consisting of both materials will be a high-contrast channel.

### 3.3. Wave Confinement

We considered buried waveguide of square cross-section in which PDMS–AAP composite form the core and PDMS clad/substrate with the 365/525 nm regulating sources assumed positioned outside but near the structure (Figure 2). We note that PDMS is transparent to UV and visible light, thus the radiations from the regulating sources reach the core with minimal attenuation. The regulating sources are analogous to the setup used for optical characterization of the PDMS–AAP composite. Then, to assess wave confinement in the waveguide, it suffices to represent the “on/off” switching of the sources with the refractive indices at trans and cis states. Wave confinement for 340 nm ultraviolent and 450 nm visible light were investigated. The wavelengths were chosen to better illustrate the system’s behavior since the refractive indices of the composite peaked at these wavelengths during reversible isomerization. High index of the composite in either of the isomeric states in the 300–450 nm wavelength range can serve the same purpose as well. However, beyond 450nm, composite index is about the same as that of pure PDMS.

As effective index method [24] translates any waveguide structure to the planar geometry, we utilize the solution to the structure in Figure 3 for a wave $E_{y}=E_{o}\left(x,y\right)e^{i\left(\beta z-\omega t\right)}$ propagating in z-direction and polarized in the y-direction. Then,
(2)$$\frac{\partial^{2}E_{y}}{\partial x^{2}}+k_{x}^{2}E_{y}=0$$
describes the transverse electric field (TE) and is solvable with different constraints. Based on empirical indices (see Figure 1), we imposed high contrast condition, $n_{core}-n_{clad}\gg 0$ and the solution:(3)$$E_{y}\left(x\right)=\left\{\begin{array}{l} \;C\;exp\left(\gamma_{2}x\right),\;x<-a\\ A\cos\left(k_{x}x+\theta \right),\;\;-a\leq x\leq a\\ \;S\;exp\left(-\gamma_{3}x\right),\;x>a \end{array}\right.$$

Here,
(4)$$k_{x}=k_{o}\sqrt{n_{core}^{2}-n_{eff}^{2}}$$
(5)$$\gamma_{2,3}=k_{o}\sqrt{n_{eff}^{2}-n_{sub,clad}^{2}}$$
and phase angle, $\theta =\left(m^{\prime }-1\right)\pi /2$. Since a waveguide is describable in terms of normalized parameters [25], it is easy to show from $E_{y}\left(x\right)$ that normalized dispersion and number of modes are respectively:(6)$$v\sqrt{1-b}=\left(m-1\right)\pi +\tan^{-1}\sqrt{b/\left(1-b\right)}+\tan^{-1}\sqrt{\left(b+\alpha \right)/\left(1-b\right)}.$$
(7)$$m^{\prime }=1+\mathrm{Int}\left[\pi^{-1}\left(v-\tan^{-1}\sqrt{\alpha }\right)\right]$$
where v is normalized frequency (*v*-number), b is relative index, and α is asymmetry factor. Both equations are vital to characterizing waveguides via analytical method on which the current work is based. The optical dispersion relation plays unique role in that *v-b* chart permits extracting the relative effective index *b* for a chosen normalized frequency of a design. With these values and empirical refractive indices of the involved materials, it is easy to simulate a waveguide of any dimension that was reduced to the slab geometry using analytical solutions and provide clear understanding of expected performance of fabricated counterpart. The fact that the range of *b* is between 0 and 1 guarantees that the result from this approach is at par with the outcome of any numerical method but with minimal computational time and is more intuitive since the analytical solution is a set of familiar functions.

We considered injected wavelength of 340 nm and three modes. Then, using the *trans* state as default, we determined design *v*-number of 7.0 and core thickness of 320 nm. The surrounding PDMS thickness is set to 160 nm on each side, and total structure length is 640 nm. Since PDMS forms the clad and substrate, the asymmetry factor is zero. The three-mode *v-b* chart is shown in Figure 4, from which we extracted the relative index b, at *v* = 7.0 in *trans* state. This approach is repeated for 450 nm signal for the same waveguide dimensions already determined with the design wavelength of 340 nm. In this case, we only need to evaluate the *v*-number for 450 nm and extract the b parameter.

To simulate the waveguide, the analytical solution, $E_{y}\left(x\right)$, was implemented on the buried waveguide via Effective index method using the geometric scheme of Figure 3b. First, the solution was applied in the y-direction for each of regions I, II, and III to determine their effective indices, N, and reduce the 3D structure to the slab geometry of Figure 3a. A second application in x-direction utilizes these indices for the final solution for the 3D waveguide. A sample set of propagation parameters for 340 nm injected signal is listed in Table 1.

For operating wavelengths of 340 and 450 nm, fixed core thickness of 320 nm and overall structure thickness and length (in z-direction) of 640 × 640 nm were used. To mimic alternate switching “on” of 365/525 nm external regulating source, we carried out the simulation with refractive indices of the core in the *trans* and *cis* states at the operating wavelengths. The result is given in Figure 5, where shaded region defines the waveguide core in the simulation. Highly confined wave is observed for the fundamental (m = 0) and first-order (m = 1) higher modes, where the evanescent components decay to zero in the clad; however, in second-order higher mode (Figure 5c), the evanescent components are very much above zero and decay slowly, which is typical of leaky confinement. So, this mode cannot be supported in a practical waveguide with same dimensions as used in the simulation. This is further emphasized by Equation (7), where $m=m^{\prime }-1$ is substituted, yielding only two supported modes, m = 0 and m = 1. Within the frame of computation and simulation, the use of $m^{\prime }=1,\;2,\ldots$ elucidates all possible modes with the quality of wave confinement, as shown in Figure 5, which is beneficial for informed practical fabrication. Now, when the core transits to the cis state through photoisomerization, as can be induced by the external regulating source, the accompanying adjustment of refractive index, which suppresses higher modes, saves a fundamental mode with reduced *v* number. For the current case, we have *v* number of 2.4, as indicated in Figure 4, and field profile as shown in Figure 5d. Evanescent waves in the latter sharply tend to zero thereby sustaining good wave confinement. In comparison with Figure 5a, Figure 5d indicates that the group velocity is higher in the cis state and the wave propagates faster. The spatial spread of guided waves for the first two allowed modes in a 640 × 640 waveguide is illustrated in Figure 6. The PDMS–AAP core presented excellent guiding ability in both isomeric states at the operating wavelength because of efficient internal reflection at core–clad and core–substrate interfaces, due to high index contrast that satisfies the condition θ_crit_ ≤ θ_i_ ≤ 90°. Where θ_crit_ is critical angle of core and θ_i_ is the incidence angle at clad–core interface. If we consider the core in trans state, Figure 3a and Snell’s law show that these angles are 54.0° and 78.2°, respectively.

Guided wave also manifested for 450 nm visible light. In this case, the waveguide permitted only single mode, the fundamental mode, in trans and cis states with *v* numbers of 1.6 and 2.0 respectively (Figure 7). Interestingly, setting the core thickness to 1.19 μm supported two modes (m = 0 and m = 1) in trans state while also permitting a second-order higher mode (m = 2) in cis state. Here, the *v* numbers are 5.90 in trans and 7.9 in cis states, a reversal of the behavior in the propagation of 340 nm in which cis state suppresses higher-order mode. 

The simulations highlight two important results: first, guided propagation is achieved in the PDMS–AAP core; and second, isomerization to cis state suppresses (permits) higher modes for ultraviolet (visible) light propagation. Consequently, single mode propagation can be obtained in cis state with appropriate choice of core thickness. It implies that one can selectively excite just the fundamental mode in an PDMS–AAP-based waveguide. This interesting characteristic may impact practical applications since multiple modes usually exist in dielectric waveguides [17] and may exist with other azo-based composites. These results indicate that in practice, the mode of propagation is tunable by changing the refractive index through photoisomerization prior to or during signal injection. Because AAP has long half-life and thermal stability, a particular isomeric state can be maintained for a significant period until otherwise stimulated. The importance of this result with regard to nanophotonics is obvious, especially in opto-mechanical and microfluidic devices utilizing optical force. The strength of the latter is dependent on the mode, while the attribute (attractive or repulsive) may be tuned by phase difference between a pair of waveguides. The in-situ regulation of index as demonstrated here can introduce a second-level regulation of the force that can potentially extend applications of the force.

### 3.4. Optical Force

Following up on the potential impact on optical force, the forces of interaction between a pair of PDMS–AAP cores in a common medium separated by distance x was investigated with respect to isomeric states (Figure 8). 

The force was described in terms of effective index [1,12] as $f=\frac{P}{c}\frac{\partial n_{eff}}{\partial x}$, which can be solved numerically. Nonetheless, because our waveguide is solvable in the first dimension, the 1-dimensional closed form [1], described as follows, is adequate for the current case:(8)$$f=\mp \frac{P}{c}\;\frac{\left(n_{c}^{2}-n_{eff}^{2}\right)\left(n_{eff}^{2}-n_{m}^{2}\right)}{(n_{c}^{2}-n_{m}^{2})\left(d+k_{o}^{-1}{\left(n_{eff}^{2}-n_{m}^{2}\right)}^{-0.5}\right)n_{eff}}\;e^{-\sigma x}$$
where *P* and *c* are power of signal in the waveguides and speed of light, respectively. The variation factor σ may be taken from closed forms for $n_{eff}$, which were variously demonstrated but share the same form [4,17]. We favor the Pernice et al. [17] interpretation, $n_{eff}\left(x\right)=n_{0}+n_{1}e^{-\sigma x}$, where $n_{o,1}$ are effective indices of isolated waveguide and coupling coefficient, respectively. Both Equation (8) and $n_{eff}\left(x\right)$ agree with the earlier stated phenomenological form. Now, a salient factor in determining σ in [12] is the tangent of transvers wavevector of free waveguide. We find $\tan\left(k_{x}\right)\approx 0$ in our case, so that the expression in [17] reduces to:(9)$$\sigma \approx 2\gamma \ln\left(2\right)$$

We substituted wavevector in the medium for the original constant since that constant tends to γ as x→∞. This is a reasonable approximation on the basis that coupling is achieved through evanescent wave. To maintain electric field profiles and parameters determined for the isolated waveguide, we reduced the core thickness to 218 nm in the surrounding medium of refractive index 1.0. 

Guided wave simulation shows that the parity of the wave alternates with mode index, i.e., m = 0 indicates even parity, while m = 1 is odd parity. Even and odd parities precipitate attractive (−)and repulsive (+) interactions, respectively [4,16]. Figure 9 illustrates the optical forces between a waveguide pair with respect to isomeric states of the core for propagated 340 nm wave.

The force is within the same order of magnitude as those of similar dielectrics in literature [4,14,17]. However, the trans–cis conformation adjusts both the strength and attribute in-situ due to change of refractive indices of the core. Force tuning is expected because of modal tuning of propagated waves in Figure 6. The full impact manifests in higher modes where, for example, 340 nm isomerization to cis state constrains all modes to a fundamental mode, leading to even parity attractive force of Figure 9c. Optical forces induce deformation in paired structures [16], which on account of the present contribution, may be regulated by photoisomerization for structure consisting of PDMS–AAP materials and similar hybrids. Furthermore, mechanical actuation was recorded in free-standing PDMS–AAP materials [22] and can enhance optical force-induced deformations. Thus, integrating the material on nanophotonic chips will provide a different level of controlled displacement.

## 4. Conclusions

We presented experimental results of in-situ reversible change of refractive index and data simulations indicating the potential of using organic photoswitching molecules within a solid host matrix, for controlling the refractive index for advanced nanophotonic applications. Considering this, we proposed a waveguide where the core is a polydimethylsiloxane–arylazopyrazole (PDMS–AAP) composite. Empirical refractive indices of the material in the trans and cis states were used to simulate buried waveguide structure. The simulation revealed well-confined waves due to the high-index-contrast and the tuning of the mode by the reversible trans-to-cis isomerization. Consequently, the calculated optical force between parallel waveguides reflected similar tuning, where the force transits from attractive to repulsive depending on the isomeric states. Our simulations suggest that harnessing the PDMS–AAP on optical chips will potentially lead to a new level of control due to in-situ change of refractive indices caused by photoisomerization of the embedded arylazopyrazole molecules.

## Figures and Tables

**Figure 1 polymers-14-00896-f001:**
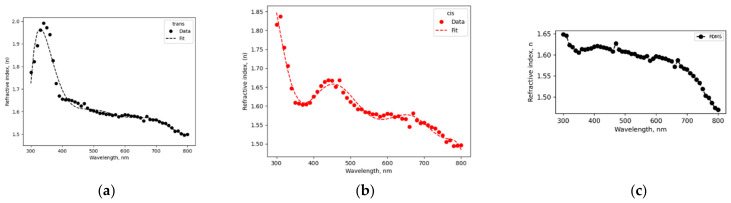
Refractive indices of polydimethylsiloxane–arylazopyrazole (PDMS–AAP) composite at (**a**) *trans* state and (**b**) *cis* state; (**c**) pure PDMS.

**Figure 2 polymers-14-00896-f002:**
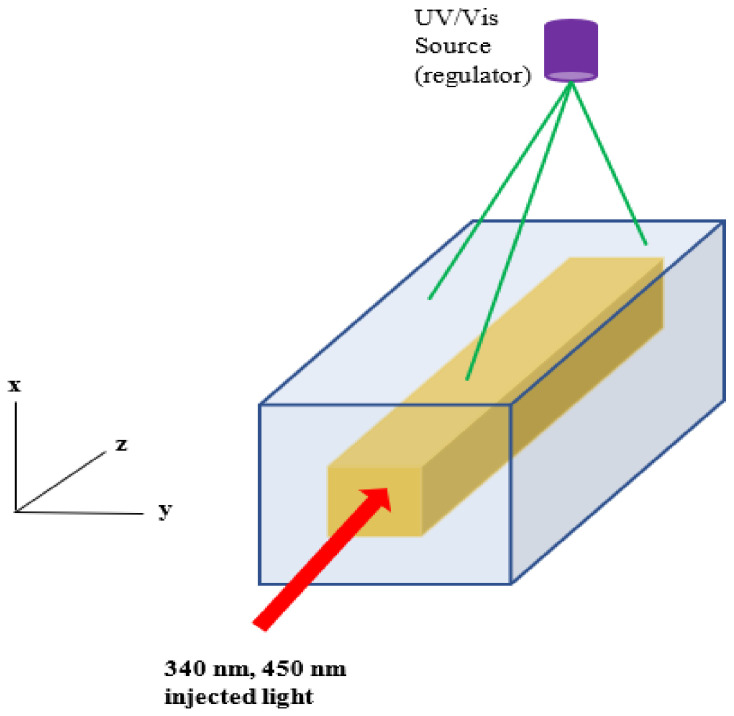
Buried waveguide structure. Gold-colored region represents PDMS–AAP composite. UV-Vis regulating source is assumed external and in proximity to waveguide.

**Figure 3 polymers-14-00896-f003:**
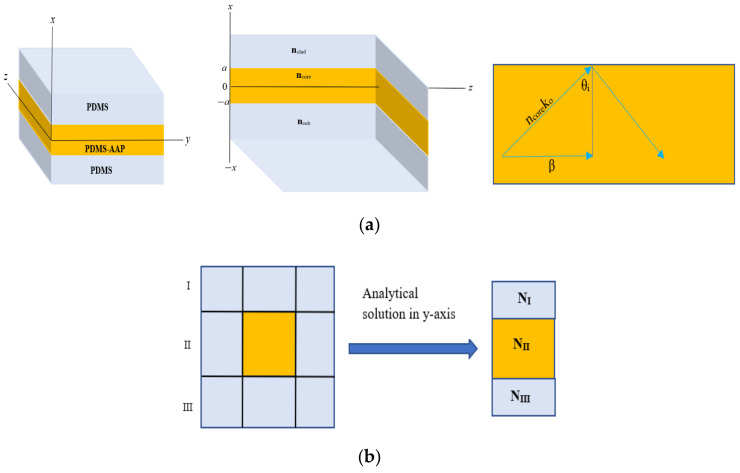
(**a**) Planar waveguide geometry. n is refractive index of clad, core, and substrate. (**b**) Effective index scheme implemented to simulated electric filed profile.

**Figure 4 polymers-14-00896-f004:**
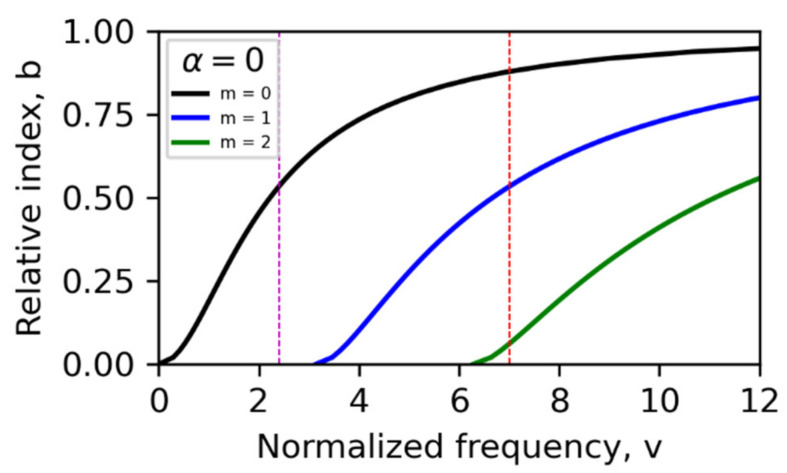
Relative index vs. normalized frequency chart for three modes for 320 nm-thick core in *trans* state. *Trans* state is design default state for operating wavelengths 340 and 450 nm. *b* values are marked by intersections of vertical line at *v* = 7.0 with mode charts.

**Figure 5 polymers-14-00896-f005:**
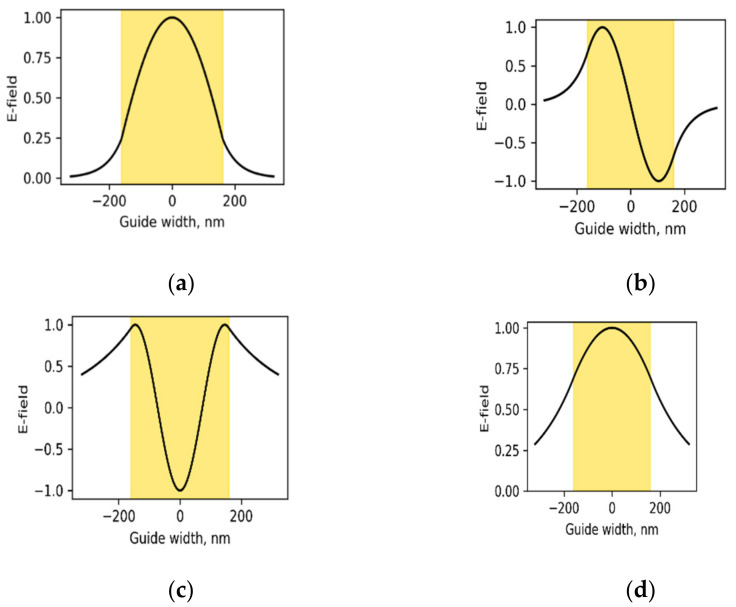
Normalized transverse electric field (TE_m_) profiles in PDMS/PDMS–AAP/PDMS buried waveguide for 340 nm wavelength in trans state of core: (**a**) fundamental mode, m = 0, (**b**) m = 1 and (**c**) m = 2. Cis state: (**d**) m = 0, all modes shift down to this fundamental mode with normalized wave frequency of 2.4. Shaded region is waveguide core.

**Figure 6 polymers-14-00896-f006:**
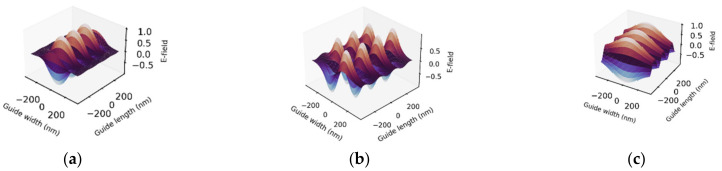
Spatial distribution of TE field in PDMS/PDMS–AAP/PDMS buried waveguide for λ = 340 nm. (**a**,**b**) m = 0, 1, respectively, with core in trans state. (**c**) m = 0, core in cis state; all higher modes shift down to this mode due to change in refractive index induced by photoisomerization.

**Figure 7 polymers-14-00896-f007:**
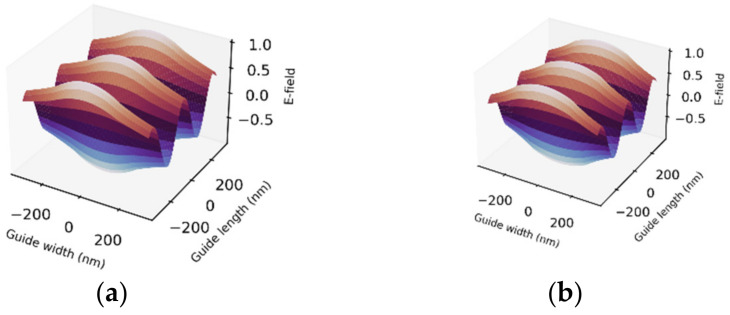
3D normalized electric field profiles in PDMS–AAP buried waveguide of λ = 450 nm. Trans (**a**) and cis (**b**) support only single mode in 320 nm-thick core. Field remained invariant, but *v* number scaled up to 2.0 in cis state.

**Figure 8 polymers-14-00896-f008:**
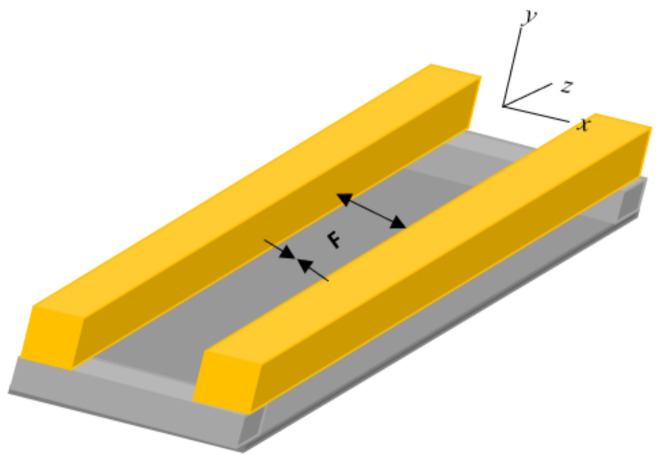
PDMS–AAP waveguide pair coupled via evanescent field in intervening space. Arrow heads’ directions indicate attractive and repulsive forces of even and odd parity modes.

**Figure 9 polymers-14-00896-f009:**
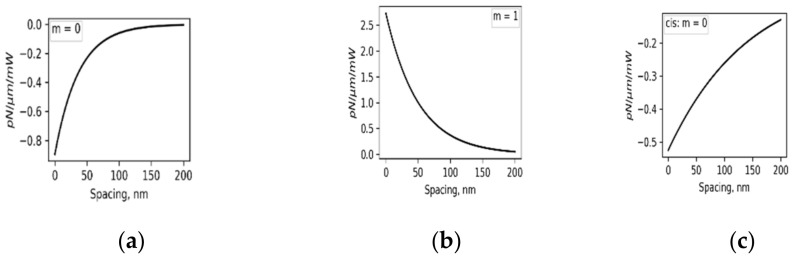
Waveguide pair interaction forces for propagating 340 nm wave. In trans state: (**a**) m = 0, even parity attractive force, and (**b**) m = 2, odd parity repulsive force. In cis state: (**c**) m = 0 supported only. Quality of force is significantly moderated by trans and cis states via changes in refractive indices of core.

**Table 1 polymers-14-00896-t001:** Propagation parameters for guided 340 nm wave in PDMS–AAP waveguide of core thickness 320 nm.

Parameter	Trans State	Cis State
n_core_	2.00	1.65
n_clad_	1.60	1.60
m′	1, 2, 3	1
* v	7.0	2.4
* b	0.885, 0.541, 0.06	0.544
* k	0.008, 0.015, 0.021	0.005
* γ	0.020, 0.016, 0.005	0.005

* Corresponding to each mode number m′.

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
