# Peer review of "A Study of Wave Confinement and Optical Force in Polydimethlysiloxane–Arylazopyrazole Composite for Photonic Applications"

_polymers, 2022, doi:10.3390/polym14050896_

Round 1
Reviewer 1 Report
This paper proposes A research of wave confinement and optical force in polydimethylsiloxane-Arylazopyrazole composite for photonic applications. This is interesting research, the following points are suggested for the author's reference.
- Why did the study only choose 340 nm and 450 nm wavelengths?
- Lines 130-135 need to be included in 3-5 years of relevant research and explain the differences with this article.
- In Figure 6 and 7. The resolution of the graphics needs to be improved, and the text has been distorted and needs to be improved in the quality of articles.
- Inconsistent text fonts in Figure 9.
- The summary section suggests revisions, with particular emphasis on the contributions of this paper and illustrating the results of simulations and actual tests.
Reviewer 2 Report
Reviewer Report
In this manuscript, the authors report on the waveguide consisting of PDMS-AAP core where the index may be regulated in-situ by alternate exposure to UV and visible light. Authors utilized buried geometry in which the core is surrounded by PDMS clad known to be transparent to UV and visible lights. A Semi-empirical simulation of electric field, in transverse electric field (TE) mode, of a signal injected into the core is carried out via a program to examine wave confinement and the effect of changing the index in-place.
Since the manuscript reports interesting and important results in the field, my recommendation is to accept it for publication in Polymers Journal, subject to the following minor revision point:
- Authors should discuss similarity and differences between graphs shown in Fig. 5(a) and Fig. 5(d), in context of trans and cis states.
- Line 321: “where ??,1 is index … respectively.” This looks incomplete, please check it.
- 6: units of guided length and width should be specified (nm).
- It would be interesting if authors could mention some relevant works which investigated the various methods for designing a different refractive index distributions in waveguides, such as:
- Takahashi et al., Optimum refractive-index profile of the graded-index polymer optical fiber, toward gigabit data links, Appl. Opt. 35, 2048-2053 (1996)
- Takahashi et al., Index profile design for high-bandwidth W-shaped plastic optical fiber, Journal of Lightwave Technology 24, 2867-2876 (2006).
- Simovic, A. Djordjevich, B. Drljaca, S. Savovic, Power flow in W-type plastic optical fibers with graded index core distribution, Optics and Laser Technology 143, 2021, 107295 (6pp).
- Simovic, A. Djordjevich, S. Savovic, Influence of depth of intermediate layer on optical power distribution in W-type optical fibers, Applied Optics 51, 2012, pp. 4896-4901.
